## Research Article

suicide; stigmatization; discrimination; support

**Corresponding author:**
Yang Jae Lee;
Email: leeyjae@uw.edu

# Experiences of adults who survived suicide attempts in rural Uganda: Stigma, support systems and reintegration

Yang Jae Lee[1,2] , Mildred Asasira[2], Lily Chang[3], Bobbi Scott[4], Alyssa Krause[5], Mildred Nakamanya[2], Van Chung[6], Rauben Kazungu[2], Ibrahim Ssekalo[2], Robert Rosenheck[7], Patrick J Raue[1], John C Fortney[1], Jürgen Unützer[1] and Alexander C. Tsai[8,9,10]

[1]Department of Psychiatry and Behavioral Sciences, University of Washington School of Medicine, USA; [2]Empower Through Health, Uganda; [3]Virginia Commonwealth University School of Medicine, USA; [4]University of Michigan-Ann Arbor, USA; [5]The Ohio State University, USA; [6]Virginia Commonwealth University, USA; [7]Yale University, USA; [8]Massachusetts General Hospital, USA; [9]Harvard Medical School, USA and [10]Mbarara University of Science and Technology, Mbarara, Uganda

## Abstract

Suicide disproportionately burdens low- and middle-income countries. In Uganda, attempt survivors encounter intense stigma, minimal mental-health services and social exclusion, elevating their risk of future attempts. Rural African data on post-attempt experiences are scarce. From June to August 2023, we conducted semi-structured, in-depth interviews in Buyende District with 18 attempt survivors, 17 relatives, 10 healthcare workers and 9 community health workers. Transcripts were translated into English and thematically analyzed using the framework method within a phenomenologically informed qualitative design. Three interlinked themes emerged. (1) Stigma-shaped immediate responses: cultural, religious and legal norms fostered moral judgment, social distancing, bureaucratic delays and occasionally police involvement. (2) Informal, uneven support: survivors relied on family aid, religious counseling and ad-hoc community advocacy; effectiveness varied widely. (3) Conditional reintegration: sustained practical help, employment and communal acceptance promoted recovery, whereas their absence perpetuated economic hardship and marginalization. Post-attempt trajectories in rural Uganda are governed by multilevel stigma and fragile support systems. Priority actions include provider training, family-community psychoeducation, stigma-reduction initiatives, structured follow-up care and decriminalization of suicide to foster compassionate responses and reduce repeat attempts.

## Impact Statements

More than 720,000 people die by suicide each year, and for each death, there are ~20 nonfatal attempts – that is, ~14–15 million attempts annually. Given that ~73% of suicides occur in low- and middle-income countries (LMICs), millions of attempts (>10 million) likely occur in LMICs each year.

In Uganda, where suicide attempts remain criminalized and mental health services are scarce and centralized, recovery after an attempt unfolds at the intersection of stigma, service gaps and economic precarity. Our study contributes context-specific evidence from rural eastern Uganda to inform frontline training, community-led after-care and policy reform that can be implemented within existing systems.

Three practical insights emerge:

1. Stigma starts at the frontline. Fear-driven reactions from health staff, police and neighbors often deepen survivors' distress and delay care. This finding pinpoints an entry-level target for brief, skills-based stigma-reduction training that ministries of health can embed in standard in-service programs.
2. Informal networks keep people alive. Families, faith leaders and community volunteers are already the main sources of emotional and material help. Partnering with these existing actors – rather than importing costly specialist services – offers an immediately scalable pathway to support survivors across similar low-resource regions.
3. Reintegration is economic as much as clinical. Survivors who regained work or farming opportunities were far less likely to remain socially isolated. Integrating livelihood support into mental-health strategies can therefore amplify both health and development gains.

By translating survivors' experiences into concrete leverage points for action, our research equips Ugandan stakeholders – and counterparts across sub-Saharan Africa – with evidence to:

(i) tailor frontline training, (ii) design community-led after-care models and (iii) advocate for decriminalizing suicide so that seeking help is never a crime. Beyond regional relevance, the study reminds the global mental-health field that effective suicide prevention must extend past the hospital door, addressing stigma, social acceptance and economic resilience in tandem.

## Background

Suicide is a leading public health concern worldwide. Over 720,000 people die by suicide each year, and for each death, there are an estimated 20 nonfatal suicide attempts (World Health Organization, 2021b). Low- and middle-income countries (LMICs) bear a disproportionate share of this burden, with 73% of global suicides occurring in LMICs (World Health Organization, 2021b). Africa has a high regional suicide rate (~11.2 per 100,000 population), with especially high rates among males (about 18.0 per 100,000) (World Health Organization, 2022). These figures likely underestimate the problem, as pervasive stigma and legal prohibitions in many societies lead to underreporting of both suicides and attempts (Adinkrah, 2016). In many parts of the world, suicide is heavily stigmatized and sometimes even criminalized, making it a "secretive act" often hidden due to taboo (Mayer et al., 2020). Suicide-related stigma – negative stereotypes, prejudice and discrimination toward people with suicidal thoughts/behaviors and those bereaved by suicide – operates at multiple levels (Link et al., 2001; Corrigan and Watson, 2002b; Pescosolido and Martin, 2015): public/enacted stigma (overt discriminatory acts) (Link et al., 1987), anticipated/perceived stigma (expectation of rejection) (Mayer et al., 2020), internalized/self-stigma (self-directed shame) (Link, 1987; Corrigan and Watson, 2002a; Corrigan and Rao, 2012) and structural stigma (laws/policies) (Corrigan et al., 2004). Families may also experience courtesy stigma (Angermeyer et al., 2003).

Uganda illustrates these dynamics in a low-resource context. The nation has one national psychiatric referral hospital (Butabika) and psychiatric units in several regional referral hospitals, but services remain highly centralized, and capacity varies widely (Kaggwa et al., 2022a). According to the WHO Mental Health Atlas, Uganda has 0.09 psychiatrists and 1.76 mental health nurses per 100,000 population (total mental health workforce ~2.57/100,000), underscoring critical human resource constraints (World Health Organization, 2021a). Suicide remains criminalized in Uganda: under Section 210 of the Penal Code, attempted suicide is a misdemeanor punishable by up to 2 years in prison (Hjelmeland et al., 2012; Kitafuna, 2022). Although this law is rarely enforced, its existence reinforces societal views of suicide as a sinful or criminal act. Certain versions of culturally Christian religious beliefs predominate that condemn suicide as a grave sin (Mugisha et al., 2013), and traditional beliefs in some Ugandan communities view suicide as a spiritual taboo or "abomination." For example, among the Acholi in Northern Uganda, a suicide death is considered a severe social transgression that disrupts communal harmony, necessitating cleansing rituals to restore balance (Mugisha et al., 2018). Such cultural narratives, echoed in other African contexts (e.g., Ghana, where suicide is seen as a moral failing and a "social injury" against the community) (Asare-Doku et al., 2017), contribute to intense stigma and social exclusion of those associated with suicidal behavior.

Against this backdrop of stigma and inadequate support, individuals who survive suicide attempts occupy a particularly vulnerable and underserved status in most societies worldwide. A prior suicide attempt is one of the strongest predictors of future completed suicide (Bostwick et al., 2016), yet attempt survivors often fall through the cracks of healthcare and community support systems.

Many survivors grapple with feelings of shame, guilt and continued psychological distress after an attempt. Studies from high-income countries indicate that suicide attempt survivors frequently experience a dual burden of stigmas attached to mental illness and suicidality (Carpiniello and Pinna, 2017). At the same time, engagement with mental health services among those who survive their attempts tends to be low; even in well-resourced environments, survivors report challenges in connecting to follow-up care and often do not receive adequate support after hospital discharge (Bruffaerts et al., 2011). In Uganda and other LMICs, these issues may be even more pronounced due to cultural silence around suicide and the near absence of formal follow-up programs. While most evidence ties stigma to lower help-seeking and worse outcomes, a small literature notes that social disapproval can function as a deterrent and lead to an increase in help-seeking behavior (Wyllie et al., 2025).

There remains a dearth of research on what happens after a suicide attempt in LMICs. Globally, suicide research has largely focused on risk factors, prevention of suicide attempts or clinical outcomes, while the lived experiences of attempt survivors remain underexplored. In Uganda, virtually no published studies have documented the personal and social experiences of suicide attempt survivors in the period following the attempt. Mugisha et al. conducted focus group discussions with community leaders and community members to examine general cultural responses to suicide in northern and central Uganda (Mugisha et al., 2011; Mugisha et al., 2018); Kizza et al. conducted psychological autopsy studies among family members and close social ties of suicide decedents, also in northern Uganda (Kizza et al., 2012a; Kizza et al., 2012b; Kizza et al., 2012c). Prior Ugandan work has shown that suicide is framed in moral/spiritual terms – for example, cleansing rituals among the Acholi – and that families of decedents carry heavy burdens of stigma and grief; recent work by Knizek et al. underscores the importance of belonging and mattering in suicidal crises (Knizek et al., 2024). Our study extends this literature by centering on attempt survivors themselves by tracking what happens after the attempt in a rural, low-resource setting where formal follow-up is scarce. This represents a critical gap: understanding the post-attempt experiences of people who have survived suicide attempts is essential for developing a compassionate healthcare response, community support mechanisms, and policies that facilitate recovery and prevent subsequent attempts.

While this study was not designed to test a particular theory, our findings align with interpersonal and ecological models of suicide and recovery. Elements described in the Interpersonal Theory of Suicide – perceived burdensomeness, thwarted belongingness and acquired capability – mirror the social and relational dynamics evident in our data, including stigma, social exclusion and barriers to recovery (Van Orden et al., 2010). We situate these dynamics within a social-ecological frame spanning individuals, families and communities, and structural forces such as law, financing and workforce capacity.

## Methods

### Study site

We conducted this qualitative study in the Irundu, Kagulu and Bukutula subcounties of Buyende District, a rural area of eastern

Uganda. Buyende has a population of ~450,000 people (Uganda Bureau of Statistics, 2017) and is characterized by agrarian communities, as the majority of residents engage in subsistence farming as their primary livelihood (Uganda Bureau of Statistics, 2022). The predominant ethnic group in the Buyende district is the Basoga, and Lusoga is the main local language. Previous research in this region has highlighted the complex healthcare-seeking behaviors of local residents, including the use of both traditional and biomedical care options (Lee et al., 2019a; Lee et al., 2019b).

### Study design and data collection

We conducted a phenomenologically informed qualitative study using semi-structured, in-depth interviews (Smith, 2009). We analyzed transcripts thematically via the Framework Method, which is appropriate for multidisciplinary teams and heterogeneous samples while maintaining a sensitivity to lived experience (Gale et al., 2013). We chose one-on-one interviews rather than focus group discussions to facilitate our ability to investigate sensitive topics surrounding suicide-related experiences and to capture nuanced, context-rich information in participants' own words. Data collection took place between June and August 2023.

We interviewed a purposive sample of suicide attempt survivors ($n = 18$), their family members ($n = 17$), healthcare workers ($n = 10$) and community health workers ($n = 9$) (total $n = 54$). Eligibility in all categories was limited to adults aged 18 years of age or older who resided in Buyende District. We recruited participants through multiple pathways.

Suicide attempt survivors were first recruited through a cohort who had agreed in previous studies to be contacted for further studies or were identified by volunteer community health workers as the study progressed. The sole criterion for inclusion in this category of study participants was having a history of at least one lifetime suicide attempt. We also interviewed (separately) the survivors' family members. The primary criterion for inclusion in this category was awareness of the suicide attempt. Among the 18 survivors we interviewed, one survivor did not tell anyone about his suicide attempt, so we did not interview any of his family members. Healthcare workers and volunteer community health workers were chosen via purposive sampling from health center rosters in the area. We used purposive sampling to capture diverse perspectives directly implicated in post-attempt pathways – survivors, family members, facility-based healthcare workers and community health workers – so we could triangulate stigma, care-seeking and reintegration from multiple vantage points.

Healthcare workers were eligible if they had at least 1 year of education past secondary school (i.e., a certificate) and if they reported experience caring for at least one patient with a prior suicide attempt. Volunteer community health workers had to be a part of the Village Health Team (VHT), which is a cadre of community-based volunteer health workers in Uganda who are selected by their communities and trained to provide basic health education and referrals to formal health facilities (Mays et al., 2017). All participants needed to be capable of providing verbal consent and maintaining the ability to provide consent throughout the interview.

Research assistants with backgrounds in social science and/or public health conducted the interviews after completing training in qualitative methods and research ethics. All research assistants interviewing participants were either Ugandan master's or bachelor's students. All interviewers were fluent in both English and Lusoga (local language). For some interviews, American research assistants (either MD or bachelor level students) observed and assisted with audio recording, but did not directly participate in the interviews.

Interviewers underwent specific training tailored for this study, which included practice interviews using semi-structured interview guides developed by the research team. Training emphasized techniques to ask open-ended questions and probe gently for richer information without influencing participants' responses. The training sessions covered role-playing scenarios, emphasizing neutrality, nonverbal communication and active listening skills to encourage open-ended responses. To ensure consistency, we used standardized scripts with open-ended questions designed to elicit detailed narratives about the experiences, circumstances and emotional contexts leading up to suicide attempts. Interview guides included initial broad prompts (e.g., "Could you tell me what happened after the attempt?") followed by structured probing questions (e.g., "What was your family's reaction?" and "What was the community's reaction?"). Responses were intentionally open-ended, allowing participants to freely articulate their perspectives and experiences without restrictions. Interviewers practiced mock interviews using role-playing exercises, receiving feedback from experienced researchers to ensure adherence to open-ended questioning and sensitivity to emotional distress.

The interview process began with VHTs visiting participant homes twice: first to confirm availability and willingness to participate, and again on the interview day to verify inclusion criteria and introduce the research team. All interviews were preceded by an explanation of the participants' rights to privacy, and consent forms were signed. For those who could not read or write, a fingerprint was taken in lieu of a signature, and a literate witness cosigned. Interviews and questionnaires were conducted at the participants' place of residence or work. In cases where both the suicide attempt survivor and family members were available simultaneously, interviews were conducted in different parts of the property to maintain privacy.

The total time for each interview lasted 30–60 min. All interviews were audio-recorded using password-encrypted tablets and uploaded to a secure database accessible only by the research team. All interviews were audio-recorded, transcribed and translated into written English within 1 week of the interview. Data collection proceeded until content saturation was achieved.

### Data analysis

We used the framework method to inductively identify recurring themes (Gale et al., 2013). Following the stages described by Gale et al.: (1) familiarization, (2) coding, (3) developing an analytical framework, (4) applying the framework, (5) charting data into a framework matrix and (6) interpretation, we iteratively analyzed transcripts alongside ongoing data collection.

For familiarization, each transcript was read in full by at least two team members (at least one Ugandan and one American research assistant). Transcripts were then coded line-by-line, with recurrent concepts labeled using inductively derived codes. To develop the initial codebook, three coding teams independently reviewed early transcripts, compared inductively generated codes and reconciled overlapping or discrepant codes into a shared codebook with agreed-upon definitions and example quotations. The developing analytical framework (i.e., the organized set of codes/categories) was refined after each batch of interviews (typically after three to five interviews), with new codes incorporated when they emerged and existing codes clarified through discussion.

To apply the framework consistently, the team held regular meetings to compare coding, resolve discrepancies through discussion to consensus (updating code definitions when needed) and agree on code definitions and applications for subsequent transcripts. When questions arose regarding cultural or linguistic meaning, we consulted Ugandan researchers and, when needed, the original interviewers to ensure accurate interpretation of participants' intended meanings in context.

In the charting stage, we summarized coded data into a framework matrix, with rows representing cases (participants) and columns representing codes/categories, while retaining links to verbatim quotations. This matrix enabled systematic comparison within and across participant groups (survivors, family members, healthcare workers and community health workers) to identify convergent and divergent patterns.

For interpretation, we grouped related codes into higher-order categories and synthesized these into themes and subthemes through iterative review of the matrix and full transcripts, attending to patterns across social levels and stakeholder perspectives. Themes were refined through team discussion to ensure they were grounded in the data and internally coherent.

We selected the framework method because it supports transparent, team-based analysis and structured comparison across a multi-stakeholder dataset while maintaining linkage to participants' original accounts through the matrix.

### Ethical considerations

All research procedures received institutional review board approval from Yale University and the AIDS Support Organization in Uganda. The consent process included a detailed explanation of privacy rights and study procedures, with forms provided in both English and Lusoga. For participant protection, a mental health safety protocol was implemented. In cases where participants expressed passive suicidal ideation without intent, the research team had to be notified, and the participant was given the option to be referred to a Psychiatric Clinical Officer (PCO). The participant was also given a suicide hotline number. In cases where the participant had active thoughts with a plan, the research team stopped the interview and immediately contacted the PCO to discuss the case. During screening, two individuals endorsed active suicidal ideation; in accordance with the study safety protocol, interviews were not conducted, and both were referred for same-day evaluation by a PCO. We did not systematically record counts of potential participants who endorsed passive suicidal ideation at screening and proceeded to participate. Participants were notified that confidentiality would be broken to maintain patient safety, a family member would be notified of the urgent health concern and transport would be provided to a health center with a PCO. All data was protected through password-encrypted tablets, a secure database with restricted access, private interview locations and separation of concurrent interviews to maintain independence.

### Reflexivity and positionality

The analytic team comprised Ugandan and non-Ugandan researchers (public health, psychiatry and social science). We engaged in reflexive memoing and team discussion to consider how our identities and training might shape interpretation – especially around religion, criminalization and community norms – to avoid reproducing stigma and to make our interpretive stance explicit.

## Results

Through analysis of interviews with adults who survived suicide attempts ($n = 18$), family members ($n = 17$), healthcare workers ($n = 10$) and VHT members ($n = 9$), we identified several interconnected themes that characterize post-attempt experiences in rural Uganda. Demographic information of people who survived suicide attempts can be found in Table 1. These themes illustrate how immediate responses, stigma, support systems and long-term reintegration processes shape survivors' recovery trajectories. The findings also highlight the role of power dynamics, systemic constraints and cultural expectations in influencing these experiences. To avoid reproducing stigma and to make our analytic stance explicit, we present some quotations with brief interpretive bridges before or after each excerpt.

### Immediate responses and manifestations of stigma

The immediate aftermath of a suicide attempt was marked by medical, social and bureaucratic challenges deeply influenced by stigma. Healthcare providers' initial reactions frequently reflected

**Table 1.** Demographic information of people who survived suicide attempts

| Sex | |
|---|---|
| F | 2 |
| M | 16 |
| **Age (years)** | |
| 20–29 | 3 |
| 30–39 | 5 |
| 40–49 | 4 |
| 50–59 | 4 |
| 60–69 | 1 |
| 70 or greater | 1 |
| **Marital status** | |
| Married or cohabiting | 16 |
| Separated | 1 |
| Widowed | 1 |
| **Religion** | |
| Protestant | 8 |
| Catholic | 6 |
| Muslim | 2 |
| Pentecostal | 2 |
| **Occupation** | |
| Peasant farmer | 16 |
| Barber | 1 |
| Unemployed | 1 |
| **Education** | |
| Never attended | 4 |
| Incomplete primary | 7 |
| Complete primary | 4 |
| O-Level | 3 |

professional uncertainty and emotional discomfort, often under-pinned by stigmatizing beliefs about suicidality. Many expressed fear or hesitation when confronted with a suicide attempt for the first time. This excerpt shows professional uncertainty and antici-pated risk, shaping an initial distancing stance:

> *"In the first place, I also have to fear this person. Because if one is like, 'After all, let me die.' This person becomes like… it is… it becomes arrogant, eh? He becomes… you know, he's (unpredictable). By the time they bring this person to you, somebody who has lost interest in life, they can even bang you. Even though you are a health worker. Can finish you, you know? So to be sincere, as I can experience, I also fear this person."* Healthcare Provider, Man

We read this as a provider navigating fear in a low-resource setting with limited suicide-care training. This uncertainty manifested as emotional distancing and bureaucratic delays, reinforcing survivors' sense of neglect and isolation. Here, bureaucratic delay and scolding are experienced as public/enacted stigma at first contact with care:

> *"When we told the healthcare workers he had taken poison, they were asking too many questions and were quarreling. We told them that they should treat the patient first and quarrel later. I told them we need to save his life."* Community Health Worker, Man

Interactions with law enforcement demonstrated structural stigma stemming from criminalization. Responses ranged from punitive measures to dismissal, further complicating survivors' initial access to care. This account links personal crisis to structural stigma arising from criminalization:

> *"The police came. They wanted to take me to prison but I gave them some money. Because they were charging me with an account of attempting suicide. After the police took money from me, I said it's ok since I'm still alive."* Survivor, Man, 57 years old

Yet, legal responses were not always uniform, and it was rare that survivors were charged with a crime. By contrast, this example shows discretionary nonpunitive handling by police:

> *"So, during that time when we went to the police station, and there was nothing done, them claiming that there was no crime for someone who has tried to kill themselves. Us here, there is no way we are going to charge him of any case. It's up to you to either take him to the hospital or somewhere else for treatment. We don't have anything to charge him of that."* Survivor, Woman, 29 years old

Variable enforcement created uncertainty but sometimes enabled care. At the family and community levels, responses shaped by stigma varied widely, from protective advocacy to skepticism and gossip, significantly influencing survivors' initial experiences of support. Some neighbors intervened compassionately:

> *"His neighbors were the first to rescue him when they saw him in that condition. They took away what he was going to swallow, they tried to calm him down to just let it go."* Community Health Worker, Man

However, other survivors experienced profound social exclusion, reinforcing stigma and isolation. This excerpt reflects social withdrawal by others, consistent with public/enacted and antici-pated stigma:

> *"Because immediately after committing suicide some family members and community members hated me because they didn't even come to check on me when I was in the hospital and even when I was back at home. Most of them ignored me. So when it came to the community members, it still remained the whole story. As others came but others did not take time to check on me."* Survivor, Man, 47 years old

Stigma at the family level was deeply intertwined with shame and fear of social repercussions, sometimes framing suicide as a criminal or disciplinary issue:

> *"As a family, and potentially me myself, I was going to be depressed and stigmatized."* Family Member, Man, 52 years old

For some families, suicide attempts were perceived as a criminal or disciplinary matter, rather than a mental health crisis, with some suggesting punitive measures. Here, a suicide attempt is framed as a disciplinary rather than a clinical matter:

> *"My parents requested the chairman to take me to prison since I had suicidal ideations."* Survivor, Man, 28 years old

A punitive frame redirected the response away from psychosocial care. Community-level stigma manifested through ridicule, exclusion and supernatural explanations for suicidal behavior. This captures ridicule and exclusion – classic public stigma at the community level:

> *"When we get to know that someone has done this, for example, taking poison, the community goes around making fun of him. They discourage him from being with other people."* Community Health Worker, Man

Mockery discouraged social participation and help-seeking. In certain cases, suicide was framed through supernatural or moral explanations, reflecting traditional beliefs about misfortune and spiritual pollution:

> *"Village people sometimes tell me I was bewitched. People don't care about things that don't concern them."* Survivor, Man, 47 years old

This misfortune was also frequently perceived to extend through-out the familial clan:

> *"You would be bringing a bad omen upon your children and the clan."* Family Member, Woman

Even community healthcare workers expressed stigmatizing attitudes, affecting survivors' experiences in medical settings:

> *"You must be careful around them because of the challenges they have. They might end up killing people in the family. We shouldn't give him our time and we should start being careful around him."* Community Health Worker, Man

Overall, the immediate post-attempt responses were heavily influenced by pervasive stigma, creating complex and challenging environments for survivors attempting initial recovery.

## Support systems: Healthcare, community and religious responses

Despite the pervasiveness of stigma, various support systems emerged as crucial to survivors' recovery. Survivors engaged with multiple support systems, including religious institutions, family networks and community outreach efforts. While some received crucial assistance, others struggled with limited access to sustained care, financial instability and societal indifference. The effectiveness of support depended on whether these systems provided consistent, compassionate and practical aid rather than reinforcing stigma.

For many survivors, financial hardship was not only a precipi-tant for the attempt but also a major barrier to receiving adequate support. Some reported receiving direct financial assistance that helped them access care or rebuild their lives:

> *"Some have recommended hospitals and even some have left some money behind for support."* Survivor, Man, 31 years old

Some family members advocated to other family members to financially support survivors:

> *"I told the father let's split some land and give it to him to rent it out so he can plant some cassava so he can recover his money."* Family Member, Woman, 52 years old

Others found support through informal social networks, where individuals provided necessities such as food:

> *"One time I had no food, I faced some lady and I said "Madam, I felt like having this but as per now I have no source of income." She said, "No, whatever you feel like, come and ask for it." And right now, for sure me there is a millet bread." Survivor, Man, 31 years old*

Religious leaders played a crucial role in survivors' recovery, particularly for families who sought spiritual explanations for suicide attempts. Some survivors actively sought out religious guidance, which provided them with emotional reassurance and conflict resolution:

> *"I went and talked to the church leaders about my problems and told them as well about killing myself but for sure they were of great importance to me because they always advised me not to kill myself or my mum but instead they called for conflict resolution and they promised to call my mum over." Survivor, Man, 36 years old*

A healthcare worker recommended survivors to seek religious leaders, as they offered an alternative form of healing, particularly for families who sought spiritual explanations for suicide attempts:

> *"I lead them to the spiritual because it was more of the spiritual intervention. They needed that kind of counseling and all that." Healthcare Worker, Man*

Social support was most effective when it included both emotional connection and practical interventions. One family member described how consistent counseling and conversation played a role in rebuilding a survivor's sense of self-worth:

> *"We did not leave him alone. We kept talking to him. And slowly, he started accepting himself again." Family Member, Woman, 43 years old*

In some cases, local leaders and community members stepped in, offering counseling and mediation to resolve family tensions:

> *"No, okay the chairman getting involved he asked her what happened,when they told him it was a family issue (domestic violence) he counseled them even right now she's here and alive."* Community Health Worker, Man

Survivors also felt relief when they had someone to confide in, even if they did not receive material support:

> *"I'm a little bit relieved because I have shared it with someone." Survivor, Man, 31 years old*

Survivors encountered diverse forms of support, including financial assistance, religious guidance, emotional reassurance and community-based interventions. While some benefitted from these resources, others continued to struggle with economic instability and a lack of long-term engagement. The effectiveness of support depended on its consistency, practicality and ability to foster meaningful recovery rather than passively offering sympathy or reinforcing stigma.

## Pathways to reintegration: Social and economic recovery after a suicide attempt

Reintegration after a suicide attempt was not automatic. Survivors had to reestablish financial stability and social relationships to regain acceptance. Those who found work, financial support or community engagement had a smoother reintegration process, while others faced ongoing economic hardship, social exclusion and a lack of long-term support.

Economic independence played a major role in reintegration. Survivors who secured employment or financial assistance were more likely to be accepted. Visible productivity signaled recovery and reshaped community perception:

> *"Right now, he grows his own food and actually in April, he had his chicken. There were really many. He sold them. He no longer behaves in a way that people will think he might want to steal their things. Right now, he's a calm person." Community Health Worker, Woman*

Economic participation expedited social acceptance. Social reintegration required active engagement from survivors, often in the face of initial community reluctance:

> *"Because he was an outgoing person, he was able to stay in the community with people. People tried to avoid him, but he did not withdraw. He kept telling them the reasons he wanted to kill himself were many, but he is sorry." Community Health Worker, Man*

Healthcare workers emphasized that recovery often required sustained support rather than just crisis intervention:

> *"Just temporarily. Temporarily he improved. But of course, these episodes keep coming. Episodes keep coming. So he needs somebody to support him continuously." Healthcare Worker, Man*

This need for ongoing support was particularly emphasized by healthcare providers who recognized that recovery is a long-term process. This reinforces a need for ongoing contact beyond initial stabilization:

> *"In all circumstances where there is need for counseling, counseling should be ongoing. It should be ongoing. You cannot say that maybe, we can now stop." Healthcare Worker, Woman*

Some VHTs played a crucial role in maintaining long-term connections with survivors:

> *"About those people, my responsibility is going to visit them and ask them how they are doing. My responsibility is to visit them because when I visit them you never know they may have something to tell me." Community Health Worker, Woman*

Many survivors continued to face challenges with family relationships and community acceptance. Family monitoring sometimes reflected ongoing concerns about survivors' stability:

> *"They have to be monitored by their family members because anything can happen. The moment this person leaves your sight, you don't know what's happening inside their brain. So they need to be monitored first of all. They need to be taken care of all the time." Healthcare Worker, Woman*

While this monitoring was intended to ensure survivors' safety, it also reflected persistent doubts about their stability, sometimes making reintegration feel conditional rather than fully restorative.

The reintegration process was closely tied to both practical achievements and social acceptance. Survivors who could demonstrate economic productivity and maintain social connections were more likely to achieve successful reintegration. However, the data suggest that ongoing support and monitoring remained crucial even after initial recovery. The findings highlight how reintegration requires not just individual recovery but the rebuilding of social and economic roles within the community.

## Discussion

Our study provides one of the few detailed explorations of the post-suicide attempt experiences in a rural African context, specifically within Buyende District of Uganda. Our results highlight critical themes: post-attempt experiences marked by pervasive stigma, varied support systems and challenges in reintegration. These

findings illustrate how survivors' recovery trajectories are intricately shaped by intersecting individual, familial, community and systemic factors. Addressing these factors is particularly crucial given that suicide attempt survivors are at significantly heightened risk for subsequent attempts.

Immediate post-attempt revealed significant uncertainty and emotional discomfort among healthcare providers, directly reflecting and reinforcing broader stigmatizing attitudes toward suicidality. This aligns with global findings that healthcare providers often feel inadequately prepared and fearful when managing suicide cases, frequently resulting in emotionally distant and judgmental interactions rooted in stigma (Jansson and Graneheim, 2018; Osafo et al., 2018; Rukundo et al., 2022). The fear expressed by healthcare workers, reflected in their emotional distancing and bureaucratic delays, highlights a need for targeted training and sensitization to ask directly about current distress, collaborate on practical next steps and arrange a timely check-in when feasible. Addressing healthcare providers' discomfort could improve the initial care experience, reducing survivors' and family members' immediate emotional distress and potentially encouraging greater healthcare utilization. The variability observed in family and community responses – ranging from compassionate advocacy to active stigmatization – is consistent with regional studies in similar African contexts, such as Ghana, where families also displayed mixed responses ranging from support to shame and ostracization (Osafo et al., 2015; Asare-Doku et al., 2017; Mugisha et al., 2018). Families in our study often oscillated between protective interventions, such as advocacy for financial and emotional support, and negative reactions, including shame, anger or even punitive measures. This dual role suggests the necessity for family-focused interventions aimed at equipping families with practical skills and compassionate strategies for managing the aftermath of suicide attempts.

Stigma emerged prominently in survivors' narratives, profoundly influencing their social integration and mental health recovery. Survivors reported extensive social exclusion driven by perceptions of moral failing, criminality and spiritual contamination – beliefs deeply rooted in religious and cultural frameworks. Similar cultural stigmatization has been documented in studies from Ghana and among the Acholi people in Uganda, where suicide attempts are viewed as profound moral and spiritual transgressions that require community cleansing (Osafo et al., 2011; Mugisha et al., 2018). Our findings align closely with this regional literature, demonstrating how stigma reinforces isolation and prolongs emotional suffering. These results suggest the importance of culturally tailored interventions aimed at stigma reduction, previously conducted successfully for severe mental illness in this setting (Lee et al., 2022; Lee et al., 2024a; Lee et al., 2024b). Additionally, although rarely enforced, the mere existence of punitive laws discourages survivors from accessing care, echoing similar issues previously reported in India before its recent decriminalization (Lew et al., 2022; Ochuku et al., 2022). Decriminalizing suicide attempts could help shift societal attitudes from punitive reactions toward supportive, reducing institutional stigma and potentially decreasing repeated attempts.

Support systems identified in our results ranged from informal community networks to religious interventions. The intermittent but impactful involvement of religious leaders and community health workers in our study underscores the potential of collaborative approaches. Formative work conducted in this region suggests that collaborative models utilizing religious leaders and traditional healers could contribute to improving mental health outcomes (Lee et al., 2024c; Lee et al., 2025a; Lee et al., 2025b). The finding of this study that religious leaders offered spiritual counseling positively impacted survivors' emotional recovery suggests that there might be an opportunity to develop interventions with religious and community leaders to create compassionate crisis responses.

Economic and social reintegration emerged as key determinants of long-term recovery. Our findings indicate that survivors who regained economic stability or community acceptance through employment or informal community support networks achieved smoother reintegration trajectories. In contrast, those lacking sustained practical or emotional support continued to experience marginalization, economic distress and persistent psychological difficulties. These results resonate with broader literature indicating that addressing mental distress in low-resource settings must address underlying socioeconomic drivers, advocating practical interventions such as vocational training, financial assistance and structured community inclusion programs (Tsai et al., 2012; Cavazos-Rehg et al., 2021).

Limitations of our study include the single-region focus and reliance on retrospective qualitative accounts, which could introduce recall bias and limit generalizability. Our sample was recruited through healthcare workers and previous research participants, leaving the representativeness of attitudes in rural Uganda recorded in this study unknown. Individuals who have not engaged with formal health systems were clearly underrepresented. The presence of both Ugandan and American researchers during interviews may also have influenced how participants discussed sensitive topics, particularly around spiritual beliefs and traditional healing practices. Our sample, 16 men and 2 women, was heavily skewed toward men. Although men are more likely to attempt suicide in Uganda (Kaggwa et al., 2022b), our data is missing potentially unique perspectives from women.

In conclusion, survivors of suicide attempts in rural Uganda face complex recovery journeys shaped by intense stigma, inconsistent support systems and significant socioeconomic reintegration challenges. The stigma deeply rooted in cultural, religious and legal frameworks isolates survivors, frequently leaving their psychological and social needs unmet. While our analysis centered on lived experience rather than types of stigma, participants' accounts contained patterns consistent with public/enacted stigma (e.g., scolding or dismissive treatment), anticipated stigma (secrecy before seeking care), internalized stigma (self-blame, feeling like a burden), courtesy stigma affecting families and structural stigma related to the legal environment. Formal mental health services are scarce, placing responsibility largely upon inconsistent informal family and community networks. To address these gaps, systemic and community-based interventions are needed. Within existing roles, first responses can emphasize nonpunitive, supportive engagement by police and health-facility staff, with attention to respectful language and rapid connection to care. Because follow-up often falls to families, explicitly inviting a supportive family member into the process (with consent) may reduce uncertainty and stigma after discharge. To keep care moving across settings, it can help to identify who is responsible next (at first contact, at the clinic and at home) and to confirm that a handoff occurred (e.g., a brief call or message) – without creating a new formal protocol. Policy reforms, notably the decriminalization of suicide attempts, alongside structured mental healthcare through trained community health workers, could lead to more effective care. Educational campaigns targeting stigma reduction, family and community-level psychoeducation and practical economic support interventions could further enhance survivors' recovery outcomes.

Ultimately, improving the post-attempt recovery process is crucial for comprehensive suicide prevention for this high-risk population. Our findings advocate for an empathetic, culturally informed and resource-sensitive approach, ensuring that survivors receive not just immediate rescue but sustained support, enabling reintegration and reducing future suicide risks.

**Open peer review.** To view the open peer review materials for this article, please visit http://doi.org/10.1017/gmh.2026.10171.

**Supplementary material.** The supplementary material for this article can be found at http://doi.org/10.1017/gmh.2026.10171.

**Data availability statement.** Data available upon reasonable request to the corresponding author.

**Author contribution.** Conceptualization: YJL and ACT. Data curation: YJL, MA, LC, BS, AK, MN, VC and RK. Formal analysis: YJL, MA, LC, BS, AK, MN, VC and RK. Funding acquisition: N/A. Investigation: YJL, MA, LC, BS, AK, MN, VC and RK. Methodology: YJL and ACT. Project administration: YJL, MA, LC, BS, AK, MN, VC and RK. Resources: YJL and RK. Software: N/A. Supervision: YJL, MA, RK, RR and ACT. Validation: YJL and ACT. Visualization: YJL. Writing – original draft: YJL, LC, BS, AK, MN and VC. Writing – review and editing: YJL, MA, LC, BS, AK, MN, VC, RK, IS, RR, PR, JF, JU and ACT.

**Financial support.** Dr. Alexander Tsai reports receiving funding from the US National Institutes of Health K24DA061696-01.

**Competing interests.** The authors declare none.

**Ethics statements.** All research procedures were approved by the institutional review boards of Yale University in the US (2000034605) and The AIDS Support Organization in Uganda (TASO-2023-222). Consistent with national guidelines, we obtained clearance to conduct the study from the Uganda National Council for Science and Technology (SS1860ES). This study was conducted in accordance with the principles of the Declaration of Helsinki. Before each interview, participants were informed of their right to privacy, the voluntary nature of participation and the specific circumstances under which confidentiality could be breached – such as in the event of acute suicidality. In such cases, the study protocol emphasized a collaborative approach: participants would first be encouraged to voluntarily involve a trusted family member or provider. If a participant declined and the risk was deemed imminent, a non-collaborative breach could be initiated in accordance with safety protocols. All participants provided written informed consent; those who could not read or write signed with a fingerprint in the presence of a cosigning witness.
A mental health safety protocol was also implemented for participant protection. In cases of passive suicidal ideation, participants were offered a referral to a PCO at the end of the interview and provided with a 24/7 toll-free number for Healthhotline, operated by the Communication for Development Foundation Uganda (CDFU). In two instances, participants expressed acute suicidal ideation. In both cases, interviews were paused and the PCO was contacted immediately to assess risk. Both participants voluntarily agreed to be evaluated by the PCO, and no involuntary or non-collaborative breaches of confidentiality were required.

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
