## [Reviewer Report]

I appreciate the opportunity to learn about the experiences of survivors, their families, and stakeholders regarding life after a suicide attempt. Thank you for providing a clear and well-structured methodology for this study and introducing the topic appropriately.

I am wondering how many participants had passive suicidal ideations and ended up participating in the study. This information will help us better understand their perspectives. Following on this, how many had active suicidal ideations and ended up not participating?

---

## [Reviewer Report]

Dear Editor,

Thank you for inviting me to review this important manuscript, and I commend the authors for engaging in this deeply sensitive and significant topic that continues to silently affect Ugandans and many individuals in low- and middle-income countries (LMICs). Studies like this give me hope and excitement to see that finally, our people are beginning to speak up and expose the silence and shame surrounding suicide.

Strengths of the Manuscript:

The article has several notable strengths. The introduction and background sections are strong and well-developed; they both integrate relevant literature on the state of suicide in Uganda with sociocultural nuances summarized effectively to make a compelling argument for the importance of this study. I commend the authors for embedding the study of suicide within these nuanced cultural and political contexts.

Under the methods section, I appreciate the authors’ choice to use an interpretive phenomenological approach to explore sensitive experiences and for selecting individual interviews over group discussions, an appropriate choice given the topic. It was also thoughtful to interchange languages (English and Lusoga) during interviews, to ensure participants’ comfort and expression. The ethical considerations section was particularly well written, reflecting deep sensitivity to the topic and careful planning for participants’ safety.

The results section is well-organized, with appropriately chosen participant quotes that effectively support each theme. My only concern is that there appears to be too much space between the quotes and the authors’ interpretive voice. I suggest minimizing that gap so that the reader can clearly see how the authors’ analytic insights connect to the participants’ experiences.

Suggestions for Improvement:

1. The paper would be strengthened by grounding the literature/analysis in an empirically validated framework for exploring suicide, such as the Interpersonal Theory of Suicide (Joiner, 2005), which posits that suicidal behavior emerges from the interaction of perceived burdensomeness, thwarted belongingness, and the acquired capability for suicide, and potentially embedding it within a cultural-ecological framework, given that ecological factors such as stigma, poor policies (e.g., criminalization of suicide), and structural barriers all complicate help-seeking for vulnerable individuals.

2. In the **impact statement**, the sentence “Each year hundreds of thousands of people in low- and middle-income countries survive a suicide attempt...” would benefit from including a more specific statistic rather than broad language.

3. The discussion section is well written, but I encourage the authors to provide more concrete recommendations in the implications section. For instance, while calls to “develop community-based interventions” or “decriminalize suicide” are valid, the paper would be stronger if the authors specify *how* these actions might begin in the Ugandan context. Consider including examples such as psychoeducation for law enforcement and healthcare workers so that suicide survivors are viewed not as criminals but as individuals needing psychosocial support.

4. It would also be helpful to include recommendations for establishing a **referral pipeline** where police, healthcare workers, and family members collaborate to ensure survivors are connected to appropriate care.

5. Finally, please consider adding implications for **clinicians**. As a therapist, I was curious to know how Ugandan therapists handle suicide cases. Are they adequately equipped to manage such situations and engage families for support? Including this perspective would enrich the practical impact of the paper.

Conclusion:

Overall, this is a strong and timely paper that sheds light on an underexplored and critical issue. With the suggested revisions, I believe it has the potential to make an important contribution to the discourse around suicide in Uganda and other LMICs. I would like to see it published once these revisions are considered.

---

## [Reviewer Report]

Thank you for the opportunity to review this extremely interesting and important manuscript. This research is important in order to highlight the impacts of suicide-related stigma and the findings are worthy of publication, I have made some minor suggestions for this paper.

Impact statement

Can you be more specific regarding the number of people who survive suicide attempt rather than saying hundreds of thousands?

Really great practical insights worthy of publication.

Introduction

The introduction is a very clear and well-structured section providing solid background on the topic. However, it would be beneficial to also highlight the possible protective nature of suicide-related stigma and how in some cases it can lead to an increase in help-seeking behaviour (recent publication by Wyllie et al., 2025). It may also be helpful to briefly elaborate on the findings of the studies mentioned at the end of the introduction section (Mugisha and Kizza).

It would be beneficial to give a definition of suicide-related stigma and to highlight the different types of suicide-related stigma within the introduction.

Methods

You state within the methods section that you used IPA so it would be important to include this within the abstract too rather than saying framework method as this could lend itself to inferences being made (e.g. use of framework analysis method) from the abstract regarding methodology. Then later in the methodology you state the use of framework method so it would be beneficial to explain the methodology regarding data analysis more clearly or use consistent language.

Would it be possible to include the full interview guide as an appendix? This would help replicability.

Results

There does not seem to be a differentiation between different types of stigma within the manuscript or participants responses, it may be useful to discuss this within the discussion section.

Tables

Why are the demographics for those who survived a suicide attempt the only ones provided? It would be useful to see the demographic information for the whole sample displayed somewhere.

---

## [Reviewer Report]

Be careful to ensure that your introduction demonstrates a nuanced understanding of the broader discourse, not merely a presentation of statistics. Consider using phrasing that reflects critical engagement with global public health realities. For example, we may open the paper with ‘’ Each year, more than 720,000 people die by suicide globally, and for every death, approximately twenty others attempt to take their own lives (World Health Organization, 2021).

It is important piece of work but please be careful with the factual accuracy of the literature and data reviewed, particularly regarding country-specific information such as Uganda’s mental health infrastructure. Given the sensitivity and relevance of this topic, it is crucial that all statistics and contextual statements are drawn from credible, up-to-date sources. This will significantly strengthen the rigor and reliability of your analysis.

Uganda exemplifies the challenges of addressing suicide in low-resource contexts. Despite growing recognition of mental health needs, the country faces significant shortages in psychiatric infrastructure and trained professionals. Uganda has one national psychiatric referral hospital (Butabika) and psychiatric units in 13 regional referral hospitals, yet most people still lack access to adequate mental health care due to stigma, distance, and underfunding Ministry of Health Uganda, 2023: WHO, 2022.

However, it’s not the only facility providing psychiatric care. Uganda also has psychiatric units in 13 regional referral hospitals, plus mental health wings in some general hospitals. Each district is meant to have at least one mental health focal person, though coverage and functionality vary widely.

According to WHO’s Mental Health Atlas (2020), Uganda had approximately:

– 1.83 mental health outpatient facilities per 100,000 population.

– 0.05 mental hospitals per 100,000 population (i.e., one large hospital).

– 0.12 psychiatric beds in general hospitals per 10,000 population.

The key challenge isn’t the absence of facilities — it’s the limited human resources and capacity. Uganda has roughly 0.08 psychiatrists per 100,000 people and 1.1 mental health nurses per 100,000 (WHO, 2022). Services remain centralized, poorly funded, and heavily stigmatized, limiting access, especially in rural and refugee-hosting areas. I see you claim to have used or followed interpretive phenomenological analysis (IPA) so we expected you or assume you prioritised deep exploration of a small number of case participants rather than large or even multi-category samples which i see in your methodology

I see you having four heterogeneous groups (survivors, family members, healthcare workers, and CHWs, totaling 54 participants this is a bit confusing and doesn’t in principle align with the core principles of IPA, which emphasizes depth over breadth.

I suggest that you consider justifying in text the larger, multi-group sample by reframing it as a phenomenologically informed qualitative design (not strict IPA).

Additionally, you make mention of purposive sampling but give no rationale for the sample composition or the logic linking these groups to the study aims for example you may consider saying we used purposive sampling to capture diverse perspectives related to suicide experiences, including survivors, family members, and front-line providers.

I quite find some of your quotes plainly presented which really seem to have some challenges. It is great to read the quotes in their verbatim form; however, given the nature of the study and the focus on stigma, I would say that studies of this kind benefit more from the author’s or researcher’s reflexivity. This approach helps ensure that the analysis does not come across as academically arrogant or as reproducing stigma, but rather critically engages with it. In qualitative research, it is not only allowed but ethically appropriate to provide slight interpretation and context for participants’ statements. Doing so helps accurately convey their meaning, situates their responses within the circumstances in which they were given, and reflects the researcher’s reflexivity

I would suggest caution in using terms like “first” when describing the novelty of the study. Such claims are very strong and require thorough evidence from the literature to support them. A more cautious phrasing such as ‘’ few studies have explored’’ Knizek, B. L., Hjelmeland, H., & Ssebunnya, J. (2024). “When you are alone you have a narrow mind, but when you are with others you have a broad mind”: A qualitative interview study of suicide attempt survivors in Uganda. International Journal of Social Psychiatry, 70(1), 58–67. https://doi.org/10.1080/17482631.2024.2424012

---

## [Editor Report]

Dear Dr. Lee, 

I have read your paper and considered the attached reviewer comments and would like to offer you the opportunity to address the reviewer comments and submit your revised manuscript. Thank you for your interest in publishing in this special issue of Global Mental Health. 

Sincerely, 

Kristin Kosyluk, Guest Editor

---

## [Reviewer Report]

I reviewed a previous version of the manuscript and so I am only commenting on my acceptance of the authors responses to my previous comments. The authors have done well to address the comments made by reviewers, however I still feel that the analysis method requires more justification and explanation. Why did you choose Framework method over reflexive thematic analysis? What were the explicit steps you took when analysing transcripts - it would be useful to follow the definitions of steps within framework analysis and then state how you met these.

---

## [Reviewer Report]

It’s good to see that the authors paid attention to all my suggestions. I am satisfied with the revisions and have no further comments.

---

## [Editor Report]

Dear Dr. Lee and Co-Authors, 

Reviewers have now had a chance to look at your responses to their comments. Reviewer 1 would like to see some further revisions/justifications regarding your approach to the data. If you are able to respond to this round of comments, we will consider a revised manuscript for the special issue on Self-harm and Suicide: A Global Priority. Thank you. 

Kristin Kosyluk

Guest Editor

---

## [Editor Report]

Dear Dr. Lee and colleagues, 

I have had a chance to review your second revision of your manuscript. You have done a nice job of justifying your use of Framework Analysis and thoroughly describing how your team addressed each step. I am pleased to inform you that we can now accept your manuscript for publication in the special issue. Thank you for your important work, which we believe makes a meaningful contribution to our understanding of the experiences of adult suicide attempt survivors in rural Uganda. 

Kristin Kosyluk

Guest Editor